# Joint Reconfiguration after Failure for Performing Emblematic Gestures in Humanoid Receptionist Robot

**DOI:** 10.3390/s23229277

**Published:** 2023-11-20

**Authors:** Wisanu Jutharee, Boonserm Kaewkamnerdpong, Thavida Maneewarn

**Affiliations:** 1Institute of Field Robotics, King Mongkut’s University of Technology Thonburi, Bangkok 10140, Thailand; wisanu.juth@kmutt.ac.th; 2Biological Engineering Program, Human Factors Engineering Research Group, Faculty of Engineering, King Mongkut’s University of Technology Thonburi, Bangkok 10140, Thailand; 3Yannix Co., Ltd., Bangkok 10150, Thailand; 4Raina Robotech Co., Ltd., Bangkok 10150, Thailand

**Keywords:** artificial bee colony algorithm, bacteria foraging optimization algorithm, bio-inspired computing, failure recovery, genetic algorithm, joint reconfiguration, humanoid robots, redundant robots

## Abstract

This study proposed a strategy for a quick fault recovery response when an actuator failure problem occurred while a humanoid robot with 7-DOF anthropomorphic arms was performing a task with upper body motion. The objective of this study was to develop an algorithm for joint reconfiguration of the receptionist robot called Namo so that the robot can still perform a set of emblematic gestures if an actuator fails or is damaged. We proposed a gesture similarity measurement to be used as an objective function and used bio-inspired artificial intelligence methods, including a genetic algorithm, a bacteria foraging optimization algorithm, and an artificial bee colony, to determine good solutions for joint reconfiguration. When an actuator fails, the failed joint will be locked at the average angle calculated from all emblematic gestures. We used grid search to determine suitable parameter sets for each method before making a comparison of their performance. The results showed that bio-inspired artificial intelligence methods could successfully suggest reconfigured gestures after joint motor failure within 1 s. After 100 repetitions, BFOA and ABC returned the best-reconfigured gestures; there was no statistical difference. However, ABC yielded more reliable reconfigured gestures; there was significantly less interquartile range among the results than BFOA. The joint reconfiguration method was demonstrated for all possible joint failure conditions. The results showed that the proposed method could determine good reconfigured gestures under given time constraints; hence, it could be used for joint failure recovery in real applications.

## 1. Introduction

Recently, a greater number of social robots have been introduced to the service industry. Anthropomorphic or humanoid social robots were designed and developed to attract special attention from human users. An anthropomorphic robot arm allows design flexible movement that can be easily adapted to various situations. A human arm can be modeled as a 7-DOF kinematic model [1]; thus, an anthropomorphic robot arm should be at least 7-DOF to be able to mimic human-like motion. An anthropomorphic robot arm that has 7-DOF would have redundancy to perform many tasks similar to humans. However, since high numbers of actuators are required for an anthropomorphic 7-DOF arm design, most humanoid robots [2] have only 6-DOF or less-than-6-DOF arms.

Since 2010, the humanoid receptionist robot called Namo, which was developed by the Institute of Field Robotics, KMUTT, has provided services for different social functions such as opening ceremonies, fashion shows, etc. [3]. Namo has two 7-DOF arms, which were designed to mimic human arms so that the robot can perform human-like upper-body movements. Over the past few years, the reliability of the robot has become a critical issue. When only one of the actuators was malfunctioning, the robot was not able to perform the tasks at hand, which caused serious problems, especially in time-critical situations. This issue raised a research question on how to allow high-degree-of-freedom humanoid robots to continue their operation even when some actuators malfunction or are damaged, especially with anthropomorphic 7-DOF arms that have redundancy in their design. The process that allows the robot system to generate a new motion trajectory for completing the tasks with the remaining actuators is called joint failure recovery.

One way to handle the problem of some actuators malfunctioning is designing and developing robots with redundancy mechanisms [4,5,6,7,8]; redundant robots are designed to have more degrees of freedom than required to perform their tasks. These redundant joints can be used to replace the failed joints and allow the robot to complete its tasks under the joint failure condition in many different ways. For example, Lewis and Maciejewski [9] used a 6 DOFs joint space PUMA560 robot manipulator for the end-effector, which requires 3 DOFs task space, and defined inverse kinematic joint velocities redistribution for the redundant joints after a joint failed and locked at a position. Elsayed et al. [10] proposed a differential-kinematics-based method to control the movement of neighbor joints of the failed joint in a snake robot in order to recover from the failure to regain the snake-like shape and resume locomotion. Chen et al. [11] proposed an optimization method to optimally control the velocity and torque of remaining joints after a locked-joint failure occurred in order to reduce the joint parameter jump of a space manipulator when carrying a load.

The concept of using redundancy in robots became commonly applied. Instead of using complex computation to reconfigure the remaining joints, another approach to handling the failure recovery problem is using soft computing techniques [12,13]. Nature provides several good examples of using adaptation to solve a problem. Animals can adapt to their injuries; although adapted gait may not be the same as normal gait, the animals can still move from one place to another. Cully et al. [14] were inspired by those animals and proposed a trial-and-error learning algorithm from the robot’s previous performance before damage occurred. Cully et al. demonstrated the algorithm performance on a hexapod robot adapting to recover from damaged legs in fire damage scenarios; the robot could adapt within two minutes. Dereli and Köker [15] used a quantum-behaved particle swarm optimization algorithm (Quantum PSO) to solve the inverse kinematic of a 7 DOFs serial robot manipulator. Compared with other swarm intelligence techniques, including particle swarm optimization (PSO), firefly algorithm (FA), and artificial bee colony (ABC), Dereli and Köker found that the estimated position of the end effector by Quantum PSO was more accurate and required less computation time. There are research studies that use different bio-inspired artificial intelligence methods for solving inverse kinematics of the 7-DOF robotics manipulator, including the slime mold algorithm [16], boundary-restricted PSO [17], firefly algorithm [18], and artificial bee colony [19].

From the literature, the strategy for robots to recover from joint failure is using redundant design and then performing computational reconfiguration of the remaining functional joints to cope with the failed joint. Inverse kinematics [20,21], quadratic programming algorithms [22], and optimization methods [11,23] are often used for joint reconfiguration; however, these methods require complex computations, which could lead to high computation resources and be time-consuming when used in real situations. The approach using metaheuristic methods or soft computing algorithms could help reduce the computational complexity and determine a good joint-reconfiguration solution for redundant robots within an acceptable computation time, as shown in [15]. Unlike in industrial robot manipulators, joint failure recovery of a humanoid receptionist robot must preserve the robot’s communication ability by its basic gestures. Gestures are a part of non-verbal communication that are important for human–robot interactive communication [24]. In this study, we proposed the use of bio-inspired artificial intelligence algorithms for the failure recovery of a service semi-humanoid robot with redundant manipulators to maintain its emblematic gestures within a given reconfiguration time constraint.

## 2. Materials and Methods

### 2.1. Namo Robot

Namo is a semi-humanoid receptionist robot, as shown in Figure 1. Namo’s head can move with three degrees of freedom. Namo has two anthropomorphic arms designed to mimic human arms so that most emblematic gestures can be naturally performed by the robot. With these two arms, Namo can perform 13 pre-defined gestures. In this study, we used the 4 main gestures for her normal operations as a receptionist robot. Figure 1 illustrates Namo’s 4 main emblematic gestures, including Wai (or Thai greeting), Bye, Salute, and Side Invite. We paid particular attention to the Wai gesture as it is the traditional greeting of Thailand and an important part of Thai social behavior.

Each arm has seven degrees of freedom. The actuators are located on Namo’s arm, as illustrated in Figure 2a. The forward kinematics of Namo’s arms were analyzed with the Denavit–Hartenberg Convention [25]. Figure 2b illustrates the kinematic chains with Denavit–Hartenberg frames of Namo’s right arm as an example. The parameter setting for the Denavit–Hartenberg convention of Namo’s arm is shown in Table 1.

### 2.2. Gesture Similarity Measurement

Unlike joint failure recovery for robot manipulators, joint failure recovery for humanoid receptionist robots requires not only the correct end-effector position but the overall robot arms’ positions must also represent the correct emblematic gestures; the output combination of elbow, wrist, and palm positions should be perceived by humans as emblematic gestures. We proposed a gesture similarity measurement to be the measure for evaluating the similarity between the new joint configuration and the original configuration for each gesture. The gesture similarity score consists of four weighted terms from four components of the arm, including the elbow, wrist, fingertip, and palm orientation, as in
score = (w_elbow_·d_elbow_) + (w_wrist_·d_wrist_) + (w_tip_·d_tip_) + (w_s_·d_s_),(1)
where d is the Euclidean distance between the 3D space position of the reconfigured joint angle and that of the reference joint angle of the gesture. Greater distance expresses less similarity to the reference gesture. The Euclidean distance is used for the elbow, wrist, and fingertip components. Figure 3b illustrates the frame-based model of Namo’s right arm in simulation, as shown in Figure 3a; the reconfigured gesture of Namo’s right arm is shown in orange, while the reference gesture is shown in blue. The orange dotted line denotes the distance vector between the joint position of the reconfigured and reference gesture at the same component.

For the hand orientation, we used the distance between rotation quaternions instead. Assuming that the reconfigured wrist is at the same position as the reference wrist, as illustrated in Figure 3c, we calculated the different angles from the distance between the rotation quaternion of the reconfigured hand, *p*, and that of the reference hand, *q*. Then, d_s_ is the angular distance between the reconfigured hand orientation and the reference hand orientation.

For Wai, both of Namo’s arms need to be in the correct position in order to represent the Thai greeting properly. Therefore, we added two more components in (1) to include the distance between the wrist of the left and right hands, d_wrist_LR_, and the distance between the fingertip of the left and right hands, d_tip_LR_, in the gesture similarity measurement, as
score = (w_elbow_·d_elbow_) + (w_wrist_·d_wrist_) + (w_tip_·d_tip_) + (w_s_·d_s_)+ (w_wrist_LR_·d_wrist_LR_ − d_offset_)+ (w_tip_LR_·d_tip_LR_ − d_offset_),(2)
where d_offset_ is the offset value for the thickness of the hand. In this study, the weight values for (1) and (2) were set as equal weight values for all components.

### 2.3. Bio-Inspired Joint Reconfiguration Method for Failure Recovery

To solve the problem of joint failure recovery for humanoid receptionist robots, the method should find the reconfigured joint angles of the remaining joints that allow the robots to perform emblematic gestures; the output gestures may not be completely the same as the reference gestures but still communicate the meanings of those gestures. Similar to the way that animals adapt themselves to walk after a leg injury, bio-inspired artificial intelligence methods simulate self-organized mechanisms occurring in nature and use them to solve complex optimization problems [26]. In this study, we proposed the use of bio-inspired artificial intelligence methods for the joint reconfiguration of humanoid receptionist robots when one reference joint angle set for each emblematic gesture is available to the robots.

Once that joint failure occurs, the robot cannot control the failed joint to a desired angle. In this study, we proposed to lock the joint at a specific angle. We determined the locked angle by averaging the joint angles of that failed joint from all reference emblematic gesture sets; the average joint angle could potentially be a less restrictive constraint to the robot system to regenerate a good gesture reconfiguration of the remaining joints for all emblematic gestures. Under the constraint that the joint angle of the failed joint is locked at a specific angle, the bio-inspired artificial intelligence methods, e.g., genetic algorithm (GA), bacteria foraging optimization algorithm (BFOA), and artificial bee colony (ABC), are used as optimizers to generate joint angle sets of the remaining joints and determine a good joint reconfiguration solution that minimizes the difference between the reconfigured gestures and the reference gestures. The similarity to the reference gesture set is the objective function for the joint failure recovery problem; the greater similarity indicates the better joint reconfigured solution. We used the proposed gesture similarity measurement in (1) as the fitness evaluation of the generated joint reconfiguration. After running iterations of bio-inspired artificial intelligence methods, the reconfigured gesture sets were obtained. Figure 4 illustrates the process of the gesture reconfiguration methods with single reference sets. Joint reconfiguration can be conducted separately for each emblematic gesture.

### 2.4. Performance Analysis

We demonstrated the performance of the bio-inspired gesture reconfiguration method with three techniques: GA, BFOA, and ABC. We explored the differences in using different techniques for the same failure recovery problem on the Namo robot. The locked joint angle for each of the joints that are averaged from the reference gesture can be expressed in Table 2. Because the performance of each algorithm depends on how the solution representation and parameter setting are set appropriately to the problem, the parameter tuning is performed using a grid search to determine the most appropriate model parameter set for each algorithm before making a comparison of the performance between methods. The grid search was carried out for the Wai gesture, which is the most complex gesture among other emblematic gestures. The tuned parameter sets given the maximum similarity gesture were then used in joint reconfiguration when each joint failed to function. The experiment setup for these algorithms is explained as follows.

#### 2.4.1. Genetic Algorithm

The GA was inspired by the evolution of life in nature. Each chromosome has genotypes that can be expressed to the phenotypes of the lifeform. Through natural selection among the population of lifeforms, better individuals can exist, mate, and reproduce new offspring. The chromosomes of the new offspring can inherit some of their parents’ genotypes through the crossover process and may have some deviated genotypes from the parents through the mutation process. With a number of generation cycles, it is possible that the offspring from the later generation can evolve to survive in the environment better than their ancestors. GA simulates such an evolution process and uses the process to solve many complex problems [27,28,29,30]. Further details about GA can be found in [27].

In this study, the genetic representation of the chromosome is set as a sequence of 6 real-valued numbers, which correspond to the 6 remaining joint angles to be reconfigured. A population of chromosomes was randomly generated with uniform distribution. The parents for crossover operations were chosen at random. We chose a one-point crossover for this problem. The population in the next generation was reproduced by generation rollover, where the new, better offspring replaced the worst individuals. The key parameters that regulate the evolution process for GA include the population size, crossover rate, mutation rate, mutation step size, and the maximum number of generation cycles.

#### 2.4.2. Bacteria Foraging Optimization Algorithm

BFOA was inspired by the foraging behavior of *E. coli* and *M. xanthus* bacteria [31]. With chemotaxis, which is the movement based on their senses of the changing concentration of nutrients in the environment, bacteria can move toward the food location; additionally, they can release chemical substances to attract and recruit other bacteria in the area to the location as well. These bacteria move with the flagella that rotate to create force and push them in one direction [32]. They can perform two movement patterns: swimming and tumbling. When they sense the target, they move with the swimming pattern toward the target. The tumbling pattern allows the bacteria to move in a random direction. Bacteria use these two movement patterns in their foraging to increase the foraging area and to collaboratively find their food; many studies [33,34] have successfully used BFOA to solve complex problems. Further details about BFOA can be found in [27].

Similar to the GA algorithm, each bacterium is a sequence of 6 real-valued numbers. The control parameters for BFOA include the population size, the number of chemotactic steps, the swimming length, the number of swim steps, the number of elimination–dispersal events, the number of reproduction steps, and the probability of elimination. Additionally, the control parameters for the swarming dynamics of the bacteria population that should be set appropriately to the application include the depth of attractant released by the cell, the width of the attractant signal, the height of the repellant effect, and the width of the repellant effect.

#### 2.4.3. Artificial Bee Colony Algorithm

ABC was inspired by the foraging behavior of honey bees [35]. During foraging, bees can be in three statuses. The bees that find food sources are called employed bees. Onlooker bees are those staying at the hive and waiting for information. The employed bees fly back to the hive and share information about the quality and location of the food sources they found through waggle dances. Scout bees are bees that explore a new food source. ABC simulates the bee foraging behavior with these three phases to find a good solution for complex problems in many research areas [35,36]. Further details about ABC can be found in [37].

The dimension of the search space in this problem is 6 according to the 6 remaining joint angles to be reconfigured. Each employed bee represents the solution or food source location in the search space. The best solution has the minimum difference from the reference gesture. The control parameters for ABC include the number of employed bees, the limit number that employed bees will search until abandoning the area to explore other areas in the search space, and the maximum number of cycles. Note that ABC uses the same number of employed bees for onlooker bees at the hive. While employed bees exploit the food source area that they found, onlooker bees can be recruited to explore the adjacent area from the food source the employed found.

## 3. Results and Discussion

In this study, we used bio-inspired artificial intelligence algorithms, including GA, BFOA, and ABC, for joint reconfiguration after a joint failure of a service semi-humanoid robot with redundant manipulators to maintain its emblematic gestures. To be used instead of using complex, time-consuming computation of inverse kinematics, the bio-inspired joint reconfiguration results must be within the acceptable difference from the reference gestures; in addition, the computation time must be low enough to allow quick maintenance so that the service robot could continue her duty. All the experiments were run on a computer with a 12th Gen Intel (R) Core (TM) i7-12700K (5 GHz) with 64 GB RAM, running on Ubuntu 22.04 with Visual Studio Code IDE, implemented with Python 3.9 and NumPy 1.20.3.

### 3.1. Parameter Set Tuning for Optimal Solution

We performed a grid search on varying control parameter values of each algorithm on the Wai gesture in order to determine a good combination of parameters that yields the minimum gesture difference from the reference gesture within 1 s of computation time for joint reconfiguration. We conducted an experiment in the case where Joint 2 failed as an example in this study. The tuned parameter sets given the maximum similarity gesture for GA, BFOA, and ABC are reported in Table 3. ABC returned the best joint reconfiguration solution that had minimum difference from the reference gesture, while the best solution from GA had the largest difference from the reference gesture. The best joint reconfiguration solution from ABC had a similar difference value to BFOA but took a little less computation time than BFOA. Although the population size for ABC seems smaller than BFOA, as shown in Table 3, which may be the reason for the lower computation time, ABC uses the same number for both employed bees and onlooker bees, so the actual population size is the same as BFOA. In terms of ease of use, ABC has a smaller number of control parameters than BFOA; hence, it seems more convenient to apply ABC in real joint reconfiguration applications.

We used the tuned parameter sets of each algorithm for emblematic gesture reconfiguration on the Namo robot when Joint 2 failed to function. The experiment was repeated 100 times in order to test if the tuned parameter sets could be used for other emblematic gestures. The resulting gesture similarity measurement to the reference gestures and resulting computation time are reported in Table 4 and Table 5, respectively.

The tuned parameter sets from performing a grid search on joint reconfiguration with the Wai gesture can be used for other gestures as well. Using the Kruskal–Wallis test, it can be concluded that using different algorithms demonstrated significantly different effects on gesture similarity and computation time for joint reconfiguration. Compared to other algorithms, after 100 repetitions, ABC resulted in the best values in many aspects. ABC yielded lower difference values from the reference gestures, except for the lowest difference in the Salute gesture; however, the values are not significantly different. ABC produced more reliable results, as shown in Table 4, with the lowest SD and interquartile range for all gestures. When comparing two unmatched algorithms’ solutions with the Mann–Whitney test, it was found that most of the algorithms’ solutions are distinctive from each other in terms of gesture similarity (*p* = 0.000), except for the Wai gesture for which BFOA and ABC did not show the difference (*p* = 0.072). On the other hand, the Mann–Whitney test on computation time suggested that all algorithms’ solutions returned with distinct computation time from other algorithms’ solutions (*p* = 0.000), except for the Salute and Side Invite gestures for which BFOA and ABC did not show the difference (*p* = 0.274 and *p* = 0.642, respectively). BFOA used the lower computation time to yield the solutions in most cases, as shown in Table 4. Nevertheless, the primary concern in choosing the algorithm for real application is the quality of the solution; the computation time is the secondary concern. Hence, the results suggested that it is possible to use ABC with the tuned parameter set for joint reconfiguration after failure.

Figure 5 illustrates the best output gestures of the four main emblematic gestures of the Namo robot in simulation from GA, BFOA, and ABC in comparison with the reference gestures. Although the output gestures from ABC are most similar to the reference gestures, the differences are not much and cannot be visually perceivable. In Table 4, the minimum difference values for the Wai and Salute gestures are higher than other gestures. From Figure 5, the output gestures from all algorithms are the same. This may be because the locked joint angle of the failed joint limits the possibility of obtaining better-reconfigured gestures.

As gradient-based optimization algorithms are also found to be efficient for solving inverse kinematics problems, we verified that the chosen optimization methods could perform better than gradient-based algorithms. We compared the performance of emblematic gesture reconfiguration when Joint 2 failed to function for the Wai gesture between using the gradient descent optimization algorithm with momentum [38] and the ABC algorithm. The tuned parameter set of the ABC algorithm used in the comparison is shown in Table 3. For the gradient descent algorithm, the decay rate was set as 0.9. The learning rate was set as 0.1 and 0.05 for comparison. Both algorithms were run with random initial seeds for 1000 iterations. Figure 6 illustrates the gesture similarity scores from gradient descent (GD) and ABC algorithms over all iterations in comparison. The best gesture similarity scores from the gradient descent algorithm with 0.1 and 0.05 learning rates are 113.95 and 112.83 mm, respectively. The best gesture similarity score from ABC is 57.16 mm. The time spent to complete 1000 iterations are 1.3550, 1.3483, and 1.0740 s for GD_0.1, GD_0.05, and ABC, respectively. Figure 6 shows that ABC can quickly converge to the solution with a better gesture similarity score than the gradient descent algorithms. The performance of gradient-based algorithms depends on the initial seed; it could lead to a local optimum. Many of the bio-inspired algorithms, including ABC, utilized a population of agents to search for the solutions collectively, so they have a greater chance of finding global optimal solutions.

### 3.2. Joint Reconfiguration for All Possible Joint Failures

It is interesting to investigate whether the tuned parameter sets from one scenario can be extended to use in other joint failure scenarios. In this experiment, we used the tuned parameter sets from Joint 2 failure to reconfigure joint angles after the failure of all possible joints. Table 6 shows the comparison of algorithm performance in terms of the minimum difference values of the output gestures from the reference gestures, the computation time used for obtaining the best joint reconfiguration, the median difference values, and the interquartile range from 100 repetitions. The best values from the comparison are shown in bold.

The results showed that the best reconfigurations, which are the reconfiguration with the minimum difference value, from BFOA and ABC, were significantly better than those from GA (Wilcoxon matched-pairs signed-rank test, *p* = 0.000 and *p* = 0.000, respectively). Between the BFOA and ABC pair groups, the Wilcoxon matched-pairs signed-rank test did not show significant differences in the minimum difference values (*p* = 0.399). When considering the algorithm performance on the reliability of the results, the Wilcoxon matched-pairs signed-rank test showed significant differences in median difference values and the interquartile range of resulting difference values between BFOA and ABC (*p* = 0.000). Hence, it may be concluded that BFOA can provide a chance to yield better joint reconfiguration, but ABC can be relied upon for determining good joint reconfiguration.

From the gesture similarity measure, the failure at one joint position could contribute to the difficulty of joint reconfiguration on each emblematic gesture differently. The best gesture difference values for the Wai gesture in Table 6 ranged the widest among the values from all gestures, from 2.2689 to 93.7706 units. Figure 7 illustrates the best reconfiguration outputs of the Wai gesture as an example. To perform the Wai gesture beautifully, the robot’s palms of both hands should be put together at the chest level. From Figure 7, the output reconfigured gestures when Joints 2 and 3 failed to have more errors than in other cases. If we use the three-dimensional Cartesian coordinate system, where the *x*-axis is in the direction of the robot’s face, the *y*-axis is pointed to the side of the robot, and the *z*-axis is pointed upward, Joint 2 is responsible for the control of the upper arm up and down to the side of the robot body (or rotating around the *x*-axis). Joint 3 is responsible for the control of the lower arm from the elbow to open or close (or rotate around the *z*-axis). When joint 2 or 3 fails and is locked at the averaged angle from all the reference gestures, the space for the robot hands to reach and join together at the chest area is limited. The joint reconfiguration method has attempted to determine a good solution with low gesture difference values, but the locked joint angle at the averaged angle from all reference gestures seemed to limit the search space. In our future study, we plan to include the locked joint angle in the problem so that the joint reconfiguration method can determine a good joint angle to lock the failed joint and may allow better-reconfigured gestures than the output in this study.

### 3.3. The Analysis of Output Gestures through the Gesture Similarity Measurement

Considering the resulting gesture similarity measurement from all possible joint failures, we found that the strategy to lock the failed joint at some specific joint angle had an immense impact on the success of performing the emblematic gestures. The average joint angle from all gestures may not always be suitable. We analyzed the distance of each component of the robot arm in all best-reconfigured gestures when each failed joint occurred. Figure 8 illustrates the boxplot of all distance components in the gesture similarity measurement in (1) and (2) for all possible joint failures. Note that the distance components of the elbow, wrist, fingertip, and palm orientation, as shown in Figure 7a–d, are calculated for all emblematic gestures in this study, but the distance components between the left and right hands, as shown in Figure 7e–f, are used for the Wai gesture only.

Joints 1–3 are located in the shoulder area and are designed to control the movement of the robot’s upper arm. Joint 1 is responsible for controlling the upper arm to swing forward and backwards. Joint 2 controls the upper arm to swing up and down to the side of the robot. Joint 3 controls the upper arm to rotate inward and outward from the robot’s body. When one of these joints fails, it affects all the components along the arm. The results in Table 6 show that the minimum difference values of the reconfigured outputs when the joints in this area failed were greater than those when other joints failed for all gestures. The boxplot in Figure 8 also confirms that when either of the joints in this group failed, the distance values between the reconfigured outputs and reference gestures of all components became high. Among the joints in this group, Joint 3 seemed to cause problems in performing the most emblematic gestures when it failed.

Although we can employ the bio-inspired AI technique to determine a good solution to control other functional joints to compensate for the failed joint that was locked to a specific angle, the compensation from other joints might not be enough to recover from the failure for performing some gestures if the locked joint is not suitable. Figure 7d shows a good example of this inappropriate locked angle for the Wai gesture. The locked angle that averages from the joint angles of all emblematic gestures causes the upper arm to rotate inward to the robot’s body. Under such limited movement, the methods determine good compensation solutions from other joints but can no longer compensate for locked joint failure. As a result, the best solution is to keep the fingertips of both hands close to each other, as shown in Figure 7d.

Joint 4 is located at the elbow and controls the movement of the lower arm upward and downward. When Joint 4 fails, the methods can determine good solutions with minimal difference values compared to when other joints fail, as shown in Table 6. Figure 8 also confirms that the distances of all components are relatively shorter than in the case where other joints failed.

Joints 5–7 are located at the wrist area and are designed to control the movement of the robot’s hand, including the palm orientation. The Wai gesture is a good example of the performance of joint reconfiguration when the joints in this area fail. Figure 7h shows good recovery from Joint 7 failure; the wrist, fingertip, and palm orientation of both hands are close, so the output gesture can indicate the Thai greeting and salutation nicely. Figure 7f illustrates that the fingertips of both hands are close to each other, but the palms are opened a little; this may be due to the locked angle of Joint 5 limiting further adjustment to close the palms. When Joint 6 fails, the locked angle limits the movement to push the palms of both hands together; Figure 7g shows that the hands are crossed in the simulation. If the robot performs this reconfigured output, the real output gesture may be different from this picture.

From the results of all possible joint failures, it can be concluded that the strategy to set the joint angle to lock the failed joint is another issue that should be considered for joint reconfiguration to recover from joint failure.

## 4. Conclusions

In this study, we proposed gesture similarity measurement for semi-humanoid robots performing emblematic gestures and proposed the use of bio-inspired AI techniques for joint reconfiguration to recover from joint failures within a given time constraint. We explored the performance of the joint reconfiguration method with a strategy that used the average values of joint angles from all gestures as the locked angle for the failed joint. We found that the joint reconfiguration method using bio-inspired AI techniques could determine good reconfigured gestures for the robot. The results suggested that ABC and BFOA have the potential to generate the best-reconfigured outputs within 1 s of computation time, which allow quick maintenance in real applications. From 100 repetitions, ABC showed the higher reliability of the results with less interquartile ranges of the results in all emblematic gestures.

By analyzing the distance between the reconfigured gestures and the reference gestures, we realized that the intuitive strategy, which uses the average values from joint angles of all emblematic gestures the robot performs as the locked angle for the failed joint, was not suitable for performing all emblematic gestures as expected. In the future, we plan to improve our joint reconfiguration method using bio-inspired AI so that the method could suggest a suitable locked angle for the failed joint as well. Nevertheless, having just one reference joint angle set for each emblematic gesture may result in difficulty in determining a suitable locked angle for all gestures. As humans, each of us performs gestures differently; we plan to explore the idea of using multiple reference joint angle sets for each gesture to aid joint failure recovery in humanoid robots performing emblematic gestures.

## Figures and Tables

**Figure 1 sensors-23-09277-f001:**
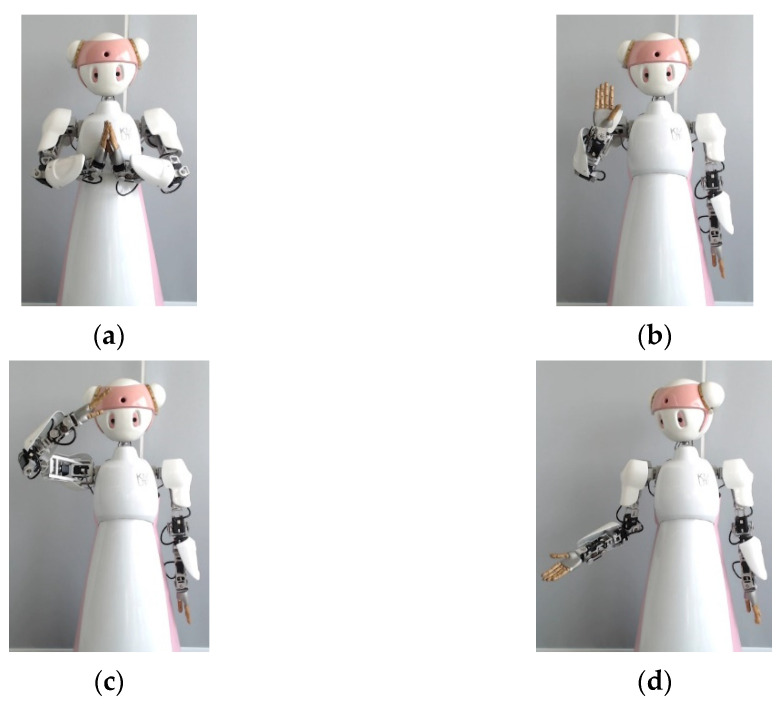
Namo’s main emblematic gestures used in this study: (**a**) Wai (or Thai greeting), (**b**) Bye, (**c**) Salute, and (**d**) Side Invite.

**Figure 2 sensors-23-09277-f002:**
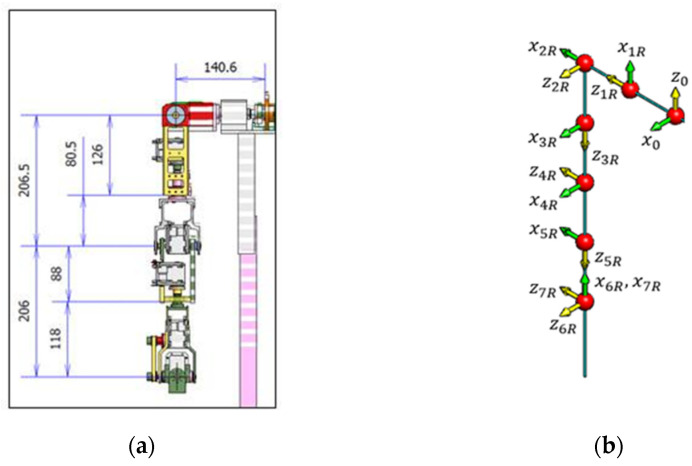
Namo’s right arm: (**a**) the joint structure; (**b**) the kinematic chains with Denavit–Hartenberg frames.

**Figure 3 sensors-23-09277-f003:**
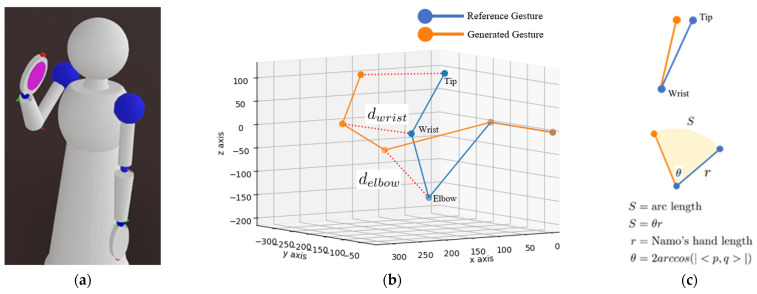
The gesture similarity measurement: (**a**) Namo’s model in simulation; (**b**) the frame-based comparison between the reconfigured gesture and the reference gesture of Namo’s right arm; (**c**) the concept for hand orientation similarity measurement.

**Figure 4 sensors-23-09277-f004:**
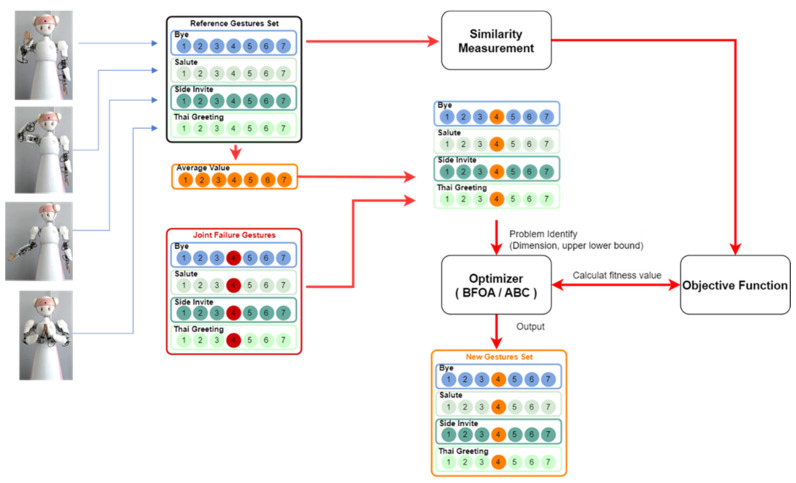
The block diagram for the proposed gesture reconfiguration method.

**Figure 5 sensors-23-09277-f005:**
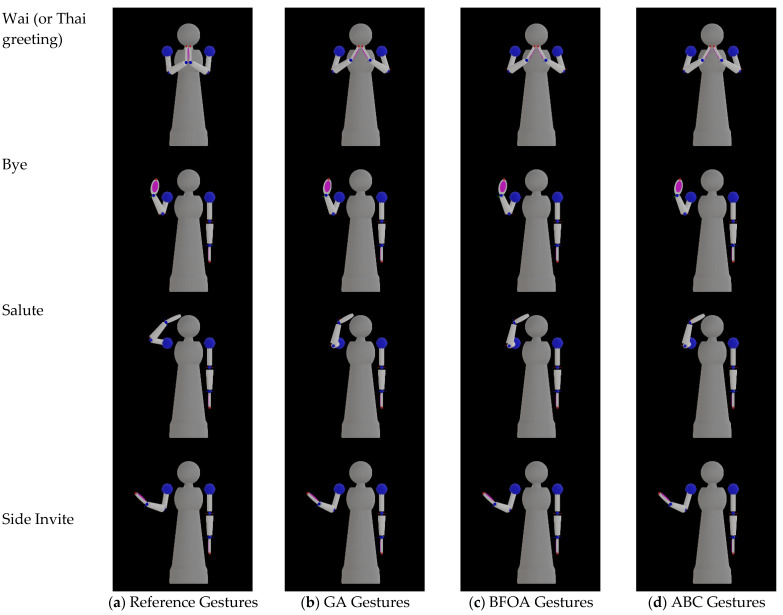
The emblematic gestures expressed by the Namo robot in simulation: (**a**) the reference gestures, (**b**) the output gestures from GA, (**c**) the output gestures from BFOA, and (**d**) the output gestures from ABC.

**Figure 6 sensors-23-09277-f006:**
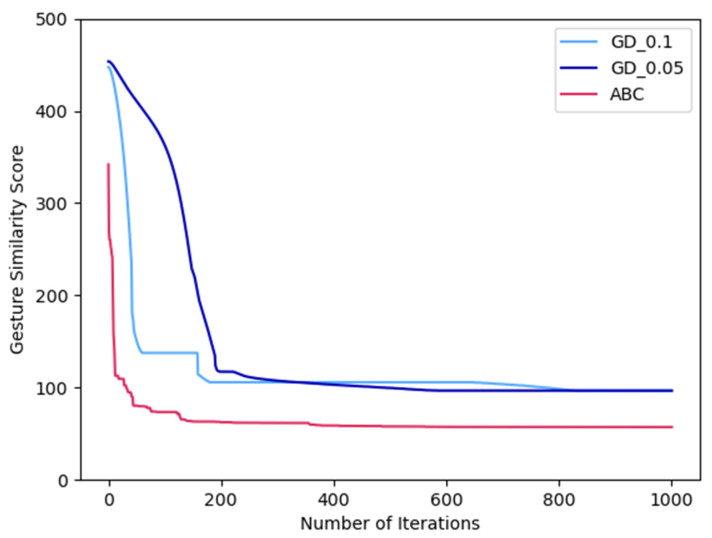
The comparison between gesture similarity scores from gradient descent (GD) and ABC algorithms over the iterations.

**Figure 7 sensors-23-09277-f007:**
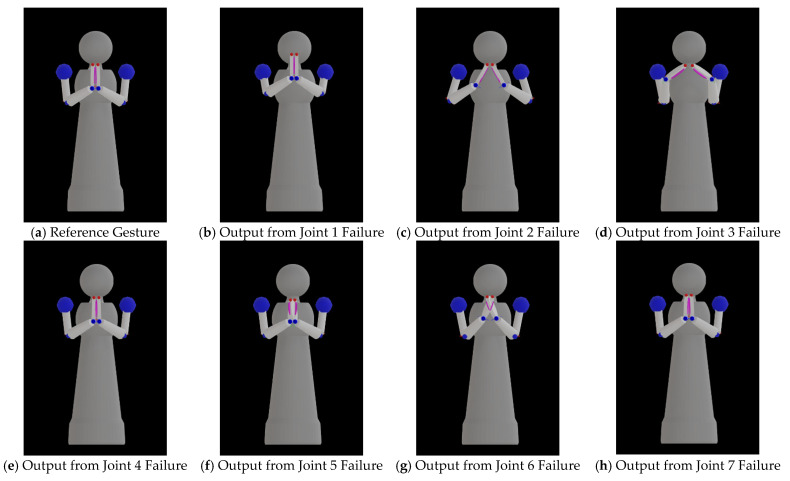
The best reconfiguration outputs of the Wai (Thai greeting) emblematic gesture in all possible joint failures on the Namo robot compared to the reference gesture: (**a**) reference gesture; (**b**) joint 1 failure; (**c**) joint 2 failure; (**d**) joint 3 failure; (**e**) joint 4 failure; (**f**) joint 5 failure; (**g**) joint 6 failure; (**h**) joint 7 failure.

**Figure 8 sensors-23-09277-f008:**
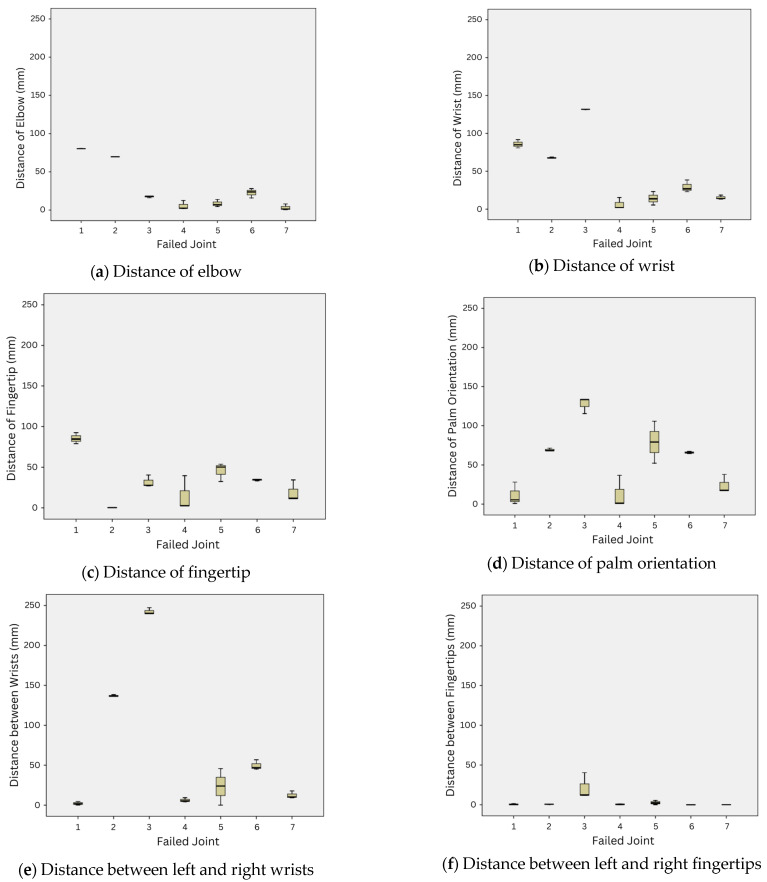
The boxplot of the distance components between the best reconfiguration outputs and the reference gestures in all possible joint failures on the Namo robot: (**a**) distance of the elbow component; (**b**) distance of the wrist component; (**c**) distance of the fingertip component; (**d**) distance of the palm orientation component; (**e**) distance between the left and right wrists for the Wai gesture; (**f**) distance between the left and right fingertips for the Wai gesture.

**Table 1 sensors-23-09277-t001:** Parameter setting for the Denavit–Hartenberg convention of Namo’s right arm.

Joint No.	$\theta_{i}$ (Degree)	$d_{i}$ (mm)	$a_{i-1}$ (mm)	$\alpha_{i-1}$ (Degree)
1	$\theta_{r1}+90{}^{\circ}$	182	0	90°
2	$\theta_{r2}+90{}^{\circ}$	0	0	90°
3	$\theta_{r3}-90{}^{\circ}$	206.5	0	−90°
4	$\theta_{r4}$	0	0	−90°
5	$\theta_{r5}+90{}^{\circ}$	206	0	90°
6	$\theta_{r6}-90{}^{\circ}$	0	0	90°
7	$\theta_{r7}$	0	0	−90°
E	0	0	−130	0

**Table 2 sensors-23-09277-t002:** The joint angles for all emblematic gestures in this study and the averaged joint angles.

Gesture	Joint Angle (Degree)						
	1	2	3	4	5	6	7
Wai	30	5	−45	90	−10	−45	45
Bye	46	−11	24	95	−53	−32	41
Salute	100	−43	−45	100	47	28	10
Side Invite	34	−8	45	72	51	26	26
**Average**	**52.5**	**−14.25**	**−5.25**	**89.25**	**8.75**	**−5.75**	**30.5**

**Table 3 sensors-23-09277-t003:** The tuned parameter sets and best joint reconfiguration solutions when joint 2 failed.

Control Parameter	Values	Best Gesture Similarity Score	Computation Time (s)
GA		58.1797	0.6540
Population size	80		
Maximum iteration	50		
Crossover rate	0.8		
Mutation rate	0.25		
Mutation step size	0.15		
BFOA		57.1479	0.5655
Population size	10		
Swimming length	0.5		
Number of elimination-dispersal events	2		
Number of reproduction steps	6		
Number of chemotactic steps	10		
Number of swim steps	15		
Probability of elimination-dispersal	0.3		
The depth of the attractant signal	0.4		
The width of the attractant signal	0.3		
The height of the repellant effect	0.4		
The width of the repellant effect	0.3		
ABC		**57.1431**	**0.5457**
Population size	5		
Maximum iteration	500		
Limit	150		

The best values are shown in bold.

**Table 4 sensors-23-09277-t004:** The descriptive statistics of difference from reference gestures of GA, BFOA, and ABC reconfigured gestures from 100 repetitions using the tuned parameter sets.

Emblematic Gestures	Descriptive Statistics of Difference from Reference Gestures						Kruskal–Wallis Test
	MIN	MAX	MEAN	SD	Median	Interquartile Range	
Wai							
GA	58.1797	88.1129	70.7241	5.9783	70.6528	9.0588	H(2) = 144.539 *p* = **0.000**
BFOA	57.1479	139.2824	64.4391	16.8914	58.4939	3.4149	
ABC	**57.1431**	**83.2653**	**59.1699**	**3.0858**	**58.1876**	**2.1447**	
Bye							
GA	6.0976	**55.3328**	26.9466	10.0887	25.6681	14.0880	H(2) = 138.612 *p* = **0.000**
BFOA	3.4758	82.5701	16.8547	17.9914	9.4605	16.5519	
ABC	**3.2491**	61.4498	**6.8030**	**6.6568**	**5.6970**	**3.7215**	
Salute							
GA	37.2962	64.8065	48.6495	6.5060	48.4423	9.7109	H(2) = 133.416 *p* = **0.000**
BFOA	35.4799	123.0403	51.0233	25.0001	37.5537	20.9101	
ABC	**35.4694**	**42.6509**	**36.2064**	**1.1113**	**35.8170**	**0.8266**	
Side Invite							
GA	8.8991	**34.1130**	19.6917	6.4367	19.4553	10.2574	H(2) = 97.740 *p* = **0.000**
BFOA	7.5411	124.6998	27.7533	28.9818	11.0276	37.2525	
ABC	**7.4523**	37.3143	**9.3562**	**3.3229**	**8.6145**	**1.8183**	

The best values are shown in bold.

**Table 5 sensors-23-09277-t005:** The descriptive statistics of computation time of GA, BFOA, and ABC output gestures reconfigured gestures from 100 repetitions using the tuned parameter sets.

Emblematic Gestures	Descriptive Statistics of Computation Time (Seconds)						Kruskal–Wallis Test
	MIN	MAX	MEAN	SD	Median	Interquartile Range	
Wai							
GA	0.6120	0.7307	0.6635	0.0316	0.6531	0.0202	H(2) = 192.905 *p* = **0.000**
BFOA	**0.4658**	0.6895	0.5669	0.0456	0.5674	0.0713	
ABC	0.5427	**0.5729**	**0.5504**	**0.0050**	**0.5500**	**0.0058**	
Bye							
GA	0.5529	0.6139	0.5725	0.0212	0.5604	0.0435	H(2) = 151.510 *p* = **0.000**
BFOA	**0.4291**	0.6526	**0.5256**	0.0371	**0.5215**	0.0478	
ABC	0.5290	**0.5550**	0.5365	**0.0045**	0.5356	**0.0037**	
Salute							
GA	0.5441	0.6098	0.5608	0.0197	0.5496	0.0402	H(2) = 88.890 *p* = **0.000**
BFOA	**0.4186**	0.6195	0.5415	0.0487	0.5433	0.0700	
ABC	0.5274	**0.5500**	**0.5356**	**0.0046**	**0.5352**	**0.0050**	
Side Invite							
GA	0.5431	0.6034	0.5601	0.0199	0.5492	0.0392	H(2) = 103.477 *p* = **0.000**
BFOA	**0.4427**	0.6216	**0.5338**	0.0394	0.5375	0.0577	
ABC	0.5261	**0.5501**	0.5356	**0.0053**	**0.5344**	**0.0077**	

The best values are shown in bold.

**Table 6 sensors-23-09277-t006:** The comparison of algorithm performance on joint reconfiguration for all joint failures using the tuned parameter sets.

Gestures/Joint Failure	Algorithm Performance											
	GA				BFOA				ABC			
	Min. Diff.	Time	Median Diff.	IQR Diff.	Min. Diff.	Time	Median Diff.	IQR Diff.	Min. Diff.	Time	Median Diff.	IQR Diff.
Wai												
Joint 1	49.1402	0.6566	69.6643	11.0471	42.8178	0.6187	50.5072	10.7610	**41.0485**	**0.5774**	**44.3499**	**2.8718**
Joint 2	58.1797	0.6540	70.6528	9.0588	57.1479	0.5655	58.4939	3.4149	**57.1431**	**0.5457**	**58.1876**	**2.1447**
Joint 3	98.6053	0.6567	107.8992	10.9072	93.7746	**0.5258**	93.9123	3.2764	**93.7706**	0.5617	**93.8543**	**1.0137**
Joint 4	18.9686	0.6536	45.7340	15.0691	2.2879	**0.5539**	14.2642	10.6975	**2.2689**	0.5570	**6.0225**	**2.7596**
Joint 5	39.3249	0.6435	55.4432	**9.6103**	**23.1424**	**0.5179**	**29.7293**	10.5429	24.2184	0.5498	30.7042	10.7042
Joint 6	35.2101	0.6485	43.5085	**8.2551**	**32.8321**	**0.5077**	36.2486	12.7388	33.0264	0.5456	**35.3097**	9.2949
Joint 7	19.5293	0.6496	39.3058	14.2708	8.8689	0.4662	12.9176	10.1463	**8.8584**	0.5688	**10.0184**	**2.8088**
Bye												
Joint 1	18.1645	0.5614	32.6537	8.6828	14.5946	0.5371	14.9520	29.3480	**14.5736**	**0.5344**	**14.8986**	**0.4006**
Joint 2	6.0976	0.5573	25.6681	14.0880	3.4758	**0.5088**	9.4605	16.5519	**3.2491**	0.5413	**5.6970**	**3.7215**
Joint 3	41.6956	0.5581	53.1471	13.9760	**38.5793**	**0.4468**	46.0145	**4.2315**	38.5830	0.5342	**39.3959**	4.2612
Joint 4	11.2942	0.6028	35.2845	17.4609	**9.5524**	0.5492	9.9890	49.1354	9.5561	**0.5338**	**9.7593**	**0.9447**
Joint 5	54.8825	0.6036	58.8346	4.7642	52.7563	**0.4211**	53.8811	**1.6746**	**52.6081**	0.5304	**53.5293**	2.2006
Joint 6	31.7665	0.5477	44.6053	10.7182	**27.5324**	**0.4849**	**28.2962**	**4.5840**	27.6241	0.5431	31.1612	6.7783
Joint 7	8.7007	0.5956	28.9101	15.9774	**7.8289**	**0.4960**	11.0214	37.2409	7.8396	0.5404	**8.9075**	**4.0756**
Salute												
Joint 1	82.3338	0.5472	84.8872	3.4310	**82.2325**	0.6137	**82.5707**	31.2031	82.2782	**0.5369**	82.7736	**0.4377**
Joint 2	37.2962	0.5469	48.4423	9.7109	35.4799	0.6014	37.5537	20.9101	**35.4694**	**0.5394**	**35.8170**	**0.8266**
Joint 3	44.2610	0.5525	57.1333	11.2424	42.0222	0.5558	43.1040	46.4772	**42.0221**	**0.5408**	**42.3335**	**0.7601**
Joint 4	21.5768	0.6080	34.7085	13.3364	18.1860	0.5824	22.5206	54.7482	**18.1818**	**0.5389**	**18.6810**	**1.8249**
Joint 5	28.2134	0.5476	38.8455	10.0302	**24.1045**	**0.4632**	**27.6639**	30.6599	24.1781	0.5323	29.4687	**5.8234**
Joint 6	34.2160	0.5542	50.9548	**11.3999**	**33.0824**	**0.5050**	35.3237	35.8533	33.2683	0.5367	**33.8175**	16.5031
Joint 7	21.3601	0.5929	30.4201	8.2655	17.4007	**0.5125**	17.5952	21.4105	**17.3955**	0.5364	**17.3971**	**0.0105**
Side Invite												
Joint 1	51.7017	0.5483	61.7445	7.1490	48.8570	0.5537	58.0899	54.1146	**48.8454**	**0.5376**	**49.1610**	**0.5528**
Joint 2	8.8991	0.5464	19.4553	10.2574	7.5411	**0.5404**	11.0276	37.2525	**7.4523**	0.5428	**8.6145**	**1.8183**
Joint 3	62.9119	0.5445	75.5241	8.7711	57.7096	**0.4627**	58.9446	20.0700	**57.7082**	0.5331	**57.7507**	**0.2847**
Joint 4	27.2899	0.5526	32.6247	6.0936	26.7073	**0.5202**	34.6626	28.3888	**26.7035**	0.5513	**26.7821**	**0.3401**
Joint 5	32.7265	0.5671	47.7179	8.7698	29.8469	**0.4813**	35.3923	18.5515	**29.6142**	0.5386	**31.0482**	**2.3620**
Joint 6	24.1722	0.5683	31.1570	8.5519	**23.7687**	**0.5164**	28.4278	42.2346	23.7868	0.5481	**26.6748**	**4.3452**
Joint 7	7.8360	0.5744	19.4936	10.9503	3.5024	0.5659	23.3782	50.3712	**3.4910**	**0.5402**	**4.4502**	**1.5727**

The best values from comparison with values in the same row are shown in bold.

## Data Availability

Data are contained within the article.

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
