# Peer review of "Joint Reconfiguration after Failure for Performing Emblematic Gestures in Humanoid Receptionist Robot"

_sensors, 2023, doi:10.3390/s23229277_

Round 1

Reviewer 1 Report

Comments and Suggestions for Authors

  1. Summary: This work seeks to maximize the similarity between human-like gestures and robot arm motions despite arbitrary joint failures. The authors use several biology inspired optimization algorithms to determine joint configurations which produce elbow, wrist, and hand positions and orientations that are similar to desired gestures.  
  2. Novelty/Contribution: The novelty of this work is based on the combination of failure recovery, human-like motion, and biology-based optimization methods.  It is common to use optimization for failure recovery, but typically gradient-based methods are used. The most important contribution this paper makes is the integration of human-like motion and failure recovery into an optimization-based framework.   
  3. Major Concerns: 
    1. The largest concern I have with this paper is the choice of optimization algorithms. Biology-based optimization methods are best when used with computationally expensive and/or extremely complex objective functions - which does not seem to be the case here. I would be interested to see how a randomly seeded gradient descent algorithm, such as BFGS, would perform comparatively. At the worst, it would demonstrate that the proposed methods are actually the most effective optimization algorithms to use.
    2. Given the large standard deviations in the results section, it appears that the optimization algorithms used were not able to consistently converge to optimal solutions. The issue here is that the paper lists runtimes of about 0.5 to 0.6 seconds for each method, but these times are only meaningful if they reflect how long it takes the optimization algorithms to converge.
    3. This paper does not have a large number of contributions. Framing the given problem as an optimization problem is a good contribution, but there is not much contributed by applying biology-based optimization to this problem. Even if biology-based optimization methods were the best choice here, it would have been more meaningful to see these optimization algorithms changed in some way to fit the specific problem at hand. 
  4.  
  5. Minor Concerns:
    1. When describing the biology-based optimization methods, it would be better to focus a little more on how these methods work computationally instead of how they function biologically. For instance, consider giving a brief description of how the crossover function for the genetic algorithm is implemented. 
    2. There were several typos and grammatical errors throughout the paper, a few I saw were:
      1. Abstract, second sentence: "When occurred with a..." should be "When occurring in a..."
      2. Section 2.3, second paragraph, third sentence: "could potentially allow the easier constraint..." should be "could potentially be a less restrictive constraint..."
    3. The final figure, figure 7, needs to be improved. The font size is too small and the axis labels should not use "code language". For example, "Failed_Joint" should just be "Failed Joint". Also, the y-axis should include units.

Comments on the Quality of English Language

Minor editing of grammar and spelling is required.

Reviewer 2 Report

Comments and Suggestions for Authors

This paper is about Joint Reconfiguration after Failure for Performing Emblematic Gestures in Humanoid Receptionist Robot. The main contribution of this paper is the proposed algorithm for joint reconfiguration of the receptionist robot called Namo under the condition that one joint actuator fails so that the robot can still perform emblematic gestures. The authors use bio-inspired artificial intelligence methods, including a genetic algorithm, a bacteria foraging optimization algorithm, and an artificial bee colony, to find good solutions for joint reconfiguration under the strategy of locking the failed joint at the average angle from all emblematic gestures.

The reviewer has  few remarks to better highlight the contribution of this article:

·        The relevance of the problem  has not been clearly stated.

·        There is no explicit mention of research gaps identified from the literature (except that you mentioned the solution based on complex computation).

·        Could you please discuss your contribution compared to the reference 17

·        In the introduction, the authors cited some examples (taken from the literature) of manipulator robots and snake robots while the robot studied in their case is a humanoid robot. I guess it makes more sense to lead a discussion with similar systems.

·        The authors underline their contribution by comparing the proposed solution to the Inverse kinematics and quadratic programming algorithms and optimization methods which are old approaches, while they dropped references dealing with  bio-inspired artificial intelligence methods.

·        Authors state that “Similar to the way that animals adapt themselves to walk after a leg injury, bio-inspired artificial intelligence methods simulate self-organized mechanisms occurring in nature and use them to solve complex optimization problems [19, 20]”. However, Reference 20 is about Time-varying sliding mode controller for heat exchanger with dragonfly algorithm!!!

·        Section 3 needs to be highlighted more. Readers need more details about the implementation (The implementation was done in which IDE? do you have videos, screenshots of the IDE...).

Comments on the Quality of English Language

No special comments

Round 2

Reviewer 1 Report

Comments and Suggestions for Authors

The authors have done a good job answering each review comment, and they have explained their choices and made some necessary changes. One important revision was emphasizing that the gesture similarity metric was developed by the authors.

However, the authors have not made a strong enough argument that the optimization methods they used are the best choice. Gradient based optimization algorithms are among the most efficient methods for solving the inverse kinematics problem [1], so their computational efficiency should not be a problem. It should be noted that (1) there are open-source implementations available (through SciPy for example) and (2) these methods use numerical differentiation - so the Jacobian of the objective function does not need to be manually derived. Additionally, the descriptions of the biology-based optimization algorithms were improved, but are not easy to understand for people without a biology background.

[1]: Beeson, Patrick and Ames, Barrett: "TRAC-IK: An Open-Source Library for Improved Solving of Generic Inverse Kinematics"

Reviewer 2 Report

Comments and Suggestions for Authors

The reviewer appreciates the work done to improve the content.